# Diagnostic Accuracy of Detective Flow Imaging Endoscopic Ultrasonography for Evaluating Blood Flow Within Mural Nodules of Intraductal Papillary Mucinous Neoplasms

**DOI:** 10.3390/diagnostics15020196

**Published:** 2025-01-16

**Authors:** Kazuki Endo, Haruo Miwa, Kazuya Sugimori, Kozue Shibasaki, Shoichiro Yonei, Yugo Ishino, Shotaro Tsunoda, Hayato Yoshimura, Akihiro Funaoka, Hiromi Tsuchiya, Ritsuko Oishi, Yuichi Suzuki, Satoshi Komiyama, Takashi Kaneko, Manabu Morimoto, Kazushi Numata, Shin Maeda

**Affiliations:** 1Gastroenterological Center, Yokohama City University Medical Center, 4-57 Urafune-cho, Minami-ku, Yokohama 232-0024, Japan; endo.kaz.bd@yokohama-cu.ac.jp (K.E.); sugimori@yokohama-cu.ac.jp (K.S.); shibasaki.koz.ah@yokohama-cu.ac.jp (K.S.); yonei.shoichiro0930@gmail.com (S.Y.); ishino.yug.cy@yokohama-cu.ac.jp (Y.I.); tsunoda.sho.au@yokohama-cu.ac.jp (S.T.); hayatotiri0913@gmail.com (H.Y.); t206071f@yokohama-cu.ac.jp (A.F.); h_tsuchiya@yokohama-cu.ac.jp (H.T.); oishi.rit.dd@yokohama-cu.ac.jp (R.O.); suzuki.yui.ar@yokohama-cu.ac.jp (Y.S.); skomiyam@yokohama-cu.ac.jp (S.K.); taku47@yokohama-cu.ac.jp (T.K.); morimoto.man.vx@yokohama-cu.ac.jp (M.M.); kz_numa@yokohama-cu.ac.jp (K.N.); 2Department of Gastroenterology, Yokohama City University Graduate School of Medicine, Yokohama 236-0004, Japan; smaeda@yokohama-cu.ac.jp

**Keywords:** detective flow imaging, intraductal papillary mucinous neoplasm, endoscopic ultrasonography

## Abstract

**Background/Objectives:** Detective flow imaging (DFI) endoscopic ultrasonography (EUS) can identify the microvascular flow imaging of a mural nodule (MN) in an intraductal papillary mucinous neoplasm (IPMN) without the use of contrast agents. This retrospective study evaluated the diagnostic accuracy of DFI-EUS and its ability to evaluate the blood flow of MNs in IPMNs. **Methods:** Between April 2021 and September 2023, 68 patients with MNs in IPMNs observed on EUS images were retrospectively analyzed. Both DFI-EUS and contrast-enhanced EUS (CE-EUS) were performed during the same session. Three expert endosonographers blinded to the patients’ clinical data assessed the MN images obtained with CE-EUS and DFI-EUS. First, DFI-EUS images were evaluated using a predefined scoring system; thereafter, CE-EUS images were evaluated. The diagnostic capability of DFI-EUS to detect MN blood flow was assessed with CE-EUS as the gold standard. Secondary outcomes included inter-reader agreement, the correlation between MN size and detection rates, and the association between DFI blood flow signal patterns and malignancy of MNs in surgically resected cases. **Results:** CE-EUS showed a contrast effect in the MN in 24 cases. Among these, DFI-EUS detected blood flow signals in 20 cases; false-positive results were not observed. DFI-EUS demonstrated a sensitivity of 83%, specificity of 100%, and accuracy of 93% for detecting MN blood flow. Inter-reader agreement was substantial (kappa values, 0.6–0.8). The subgroup analysis revealed that all MNs ≥ 10 mm had detectable blood flow on DFI-EUS, whereas MNs < 10 mm had reduced detection rates (75%; 12/16 cases). No significant correlation between the DFI blood flow signal patterns and MN malignancy of resected cases was observed. **Conclusions:** DFI-EUS demonstrated high diagnostic accuracy for detecting MN blood flow. Because of its simplicity and cost-effectiveness, DFI-EUS could be an alternative to CE-EUS for patients with MNs inside IPMNs.

## 1. Introduction

Intraductal papillary mucinous neoplasms (IPMNs) were first identified in 1982 as mucin-producing cystic neoplasms of the pancreas [1]. The frequent use of radiological diagnostic modalities, such as computed tomography (CT) and magnetic resonance imaging (MRI), and recent advancements have led to increased opportunities to incidentally diagnose IPMN [2,3,4]. Approximately 80% of incidentally identified IPMNs are branched ductal IPMNs, which are associated with the risk of malignant transformation; however, the early diagnosis of malignancy is challenging [5,6,7]. A mural nodule (MN) in an IPMN is a risk factor for malignancy [8,9,10,11,12], and its identification plays a pivotal role in clinical decision-making. The guidelines describe enhanced MNs larger than 5 mm as high-risk stigmata, thus indicating the need for surgical treatment of the IPMN [5]. A meta-analysis published in 2018 identified the MN as an independent risk factor for the malignant transformation of IPMNs, regardless of the IPMN type [13]. As a result, the early and accurate detection of MNs has become crucial in determining the treatment strategy for IPMN. Several studies have suggested that endoscopic ultrasonography (EUS) can effectively detect MNs [14]. When detecting MNs using EUS, differentiation from mucus clots can be challenging. Mucus clots are typically hypoechoic with smooth edges and a hyperechoic rim [15]. However, all three of these features are not often observed, thus making it difficult to differentiate an enhanced MN from a mucus clot without information regarding blood flow (Figure 1 and Figure 2). To address this issue, contrast-enhanced (CE) EUS can be used to assess the vascularity of MNs [16,17,18,19]. However, CE-EUS has disadvantages, including the cost of contrast agents, prolonged examination time, and risk of allergic reactions. With recent technological advancements, new imaging techniques collectively termed microvascular flow imaging (MVFI) have been developed, enabling the visualization of fine, low-velocity blood flow previously undetectable by conventional Doppler methods [20]. While the utility of MVFI has been widely demonstrated in transabdominal ultrasonography, it was not available for EUS until recently. Previously, eFLOW, a high-sensitivity Doppler technique, was the most advanced method for blood flow imaging in EUS [21]. Although effective, eFLOW has limitations in detecting low-velocity or microvascular flow patterns. This gap can be addressed with the introduction of detective flow imaging (DFI), a type of MVFI, which enables its application not only in transabdominal ultrasonography but also in EUS (DFI-EUS) [22]. Similar to Doppler imaging, DFI-EUS can be easily used without contrast agents and enables the detection of the microcirculation with greater sensitivity than that of Doppler imaging. DFI-EUS may replace CE-EUS for evaluating MN blood flow due to its non-invasive nature, high sensitivity, and simplicity. However, studies investigating its efficacy remain limited, and its diagnostic accuracy for IPMNs has yet to be fully validated [23,24]. Therefore, this study aimed to evaluate the ability of DFI-EUS to assess MN blood flow in IPMN patients, focusing on its potential as an alternative to CE-EUS.

## 2. Materials and Methods

### 2.1. Patients

Between April 2021 and September 2023, 408 patients underwent EUS for IPMN at Yokohama City University Medical Center. Of these patients, 68 with mural lesions within the IPMNs on EUS images who underwent DFI-EUS and CE-EUS were retrospectively enrolled in this study. The inclusion criteria for this study were as follows: an MN was detected by EUS; both DFI-EUS and CE-EUS were performed during the same examination; and the DFI-EUS evaluation was performed prior to CE-EUS to avoid the influence of contrast agents.

### 2.2. Definition of IPMNs

Lesions identified using abdominal ultrasound, CT, or MRI were diagnosed as IPMNs according to the appropriate criteria. Branch duct IPMNs were defined as cystic lesions with a diameter ≥5 mm that communicated with the main pancreatic duct (MPD). If direct confirmation of communication between the cyst and MPD was not possible, then cystic lesions with presumed communication based on imaging findings were also classified as branch duct IPMNs. Main duct IPMNs were defined as those with MPD dilation ≥5 mm without any other identifiable cause. Lesions exhibiting both characteristics were classified as mixed-type IPMNs.

### 2.3. Definition of Mural Nodules

During this study, an MN was defined as a protrusive lesion ≥1 mm observed within the cystic lesion and main pancreatic duct of the IPMN. An MN that exhibited enhancement on CE-EUS was defined as an enhanced MN. In contrast, an MN without enhancement was defined as a mucus clot.

### 2.4. DFI

Similar to Doppler imaging, DFI can be performed immediately by pressing a single button of the ultrasound system. During Doppler imaging, a wall filter is used to eliminate motion artifacts from the surrounding tissues; however, this has limitations in the ability to detect microcirculation. In contrast to Doppler imaging, DFI can eliminate tissue motion artifacts using an original algorithm. DFI uses a multidimensional filter to analyze motion artifacts and applies an adaptive algorithm to detect and remove tissue motion at high frame rates. As a result of this technology, DFI achieves the detection of low blood flow that is impossible to identify using Doppler imaging (Figure 3).

### 2.5. EUS Examination

EUS was performed or supervised by endoscopists with more than 5 years of EUS experience. A convex echoendoscope (GF-UCT260; Olympus, Tokyo, Japan) and an ultrasound system (ARIETTA 850; FUJIFILM Medical Co., Ltd., Tokyo, Japan) were used. All patients were examined while in the left lateral recumbent position. Sedation was primarily achieved with propofol or midazolam, and diazepam was added as needed. The blood pressure, heart rate, SpO_2_ levels, and EtCO_2_ levels were monitored during the examination. The size of the MN was measured when it was identified within the IPMN. Subsequently, DFI-EUS was performed to assess the internal blood flow of the MN, and the images were stored. The size of the region of interest (ROI) and color gain were adjusted based on the sizes of the IPMN and MN determined by each endoscopist during DFI-EUS. CE-EUS was performed using a perflubutane-based second-generation ultrasound contrast agent (Sonazoid; GE Healthcare Pharm, Tokyo, Japan) at the end of the EUS examination. The amount of Sonazoid was adjusted to 0.015 mL/kg based on the patient’s body weight and administered through a peripheral vein. After Sonazoid was injected, the MN was observed for 90 s at a mechanical index of 0.2 to 0.3, and video recordings were performed. After the examination, still images were captured at the moment of peak enhancement of the pancreatic parenchyma.

### 2.6. Image Evaluation

The assessments of the internal blood flow of the MN using DFI-EUS and CE-EUS were independently evaluated by three experts (K.S., H.M., and T.K.) with more than 10 years of EUS experience and at least 1 year of DFI-EUS experience. Clinical information and the final diagnoses were blinded. For the blinded evaluation, the most evaluable static images were stored. For CE-EUS, images that allowed evaluation, including MNs, at the peak enhancement of the pancreatic parenchyma were selected. For DFI-EUS, images without motion artifacts were selected, referencing the findings of the surrounding pancreatic parenchyma. To minimize bias, the images obtained using each modality were separated and randomized before the review process. Additionally, readers first evaluated the DFI-EUS images; thereafter, they evaluated the CE-EUS images. This approach ensured that readers were blinded to the CE-EUS findings during the initial assessment of the DFI-EUS images. The evaluations were conducted using a questionnaire that included specific criteria for both DFI-EUS and CE-EUS, and readers were asked to categorize the presence or absence of the MN internal blood flow or classify the case as “unevaluable”. Cases in which at least one of the three readers determined that pancreatic blood flow could not be evaluated were excluded from the final analysis. When the ratings of the three readers differed, the majority rating was adopted. After identifying blood flow in MNs, the signals detected on DFI-EUS were classified as either dotted or linear, and the majority opinion was used as the final classification. For surgically resected cases, these classifications were compared with the MN malignancy grades.

### 2.7. Study Design

The primary outcome of this retrospective study was the diagnostic accuracy of DFI-EUS compared to that of CE-EUS for MNs in IPMNs. The diagnostic accuracy of DFI-EUS was assessed by comparing its ability to detect vessels in MNs and that of CE-EUS, which was used as the gold standard. Diagnostic accuracy was defined as the proportion of correctly identified cases (calculated as the sum of true-positive and true-negative cases divided by the total number of evaluated cases). The evaluation focused on concordance between DFI-EUS and CE-EUS findings regarding the presence and characteristics of MNs, including their contrast enhancement patterns. Additionally, we analyzed the agreement rates of image evaluations among blinded readers, the ability to detect blood flow based on MN size, and the vascular structures of MNs classified as benign or malignant in surgical cases.

### 2.8. Statistical Analysis

Categorical variables are presented as frequencies and percentages. Continuous variables are presented as medians and ranges. All statistical analyses were performed using EZR (Saitama Medical Center, Jichi Medical University, Saitama, Japan), which is a graphical user interface for R version 3.6.2 (R Foundation for Statistical Computing, Vienna, Austria). More precisely, it is a modified version of R commander designed to add statistical functions frequently used for biostatistics [25]. A comparison of the ability of DFI-EUS and that of CE-EUS to detect blood signals in MNs was performed using McNemar’s test. Inter-reader agreement was evaluated using Landis and Koch’s proposals (Cohen’s kappa value < 0, no agreement; 0–0.20, slight agreement; 0.21–0.40, fair agreement; 0.41–0.60, moderate agreement; 0.61–0.80, substantial agreement; and 0.81–1, high agreement). Qualitative variables were used to compare categorical variables using Fisher’s exact test. Statistical significance was set at *p* < 0.05.

### 2.9. Ethics

This study was approved by the Yokohama City University Ethics Committee (no. F220900020) and adhered to the ethical standards outlined by the Institutional Research Committee and the latest Declaration of Helsinki. The requirement for patient consent was not required, and information was obtained through an opt-out process because this was a retrospective study. All authors had complete access to the study data and assumed final responsibility for the decision to submit the manuscript for publication.

## 3. Results

### 3.1. Patient Characteristics

Among the 68 patients included in this study, 46 were diagnosed with branch duct IPMNs (BD-IPMNs), accounting for 68% of cases. Six patients (9%) had main duct IPMNs (MD-IPMNs), and sixteen patients (23%) presented with mixed-type IPMNs (Table 1).

The median age of the patients was 74 years (range: 50–87 years), with 38 males (56%) and 30 females (44%). The median diameter of the main pancreatic duct (MPD) was 3.1 mm (range: 1–23 mm). The maximum diameter of the cysts containing mural nodules (MNs) had a median of 23 mm (range: 6–76 mm). MNs were most frequently observed in the pancreatic head, accounting for 59% (40 cases). This was followed by the pancreatic body with 23% (16 cases) and the pancreatic tail with 18% (12 cases). Similarly, the maximum diameter of the MNs themselves had a median of 5.0 mm (range: 1.7–30 mm).

### 3.2. Proportion of Cases Evaluated Using CE-EUS and DFI-EUS

Among 68 cases, pancreatic blood flow images of eight cases obtained using DFI-EUS were difficult to evaluate; however, all images of cases obtained using CE-EUS were evaluated (88% [60/68] vs. 100% [68/68]; *p* = 0.006). Difficulty performing evaluations of images obtained with DFI-EUS was attributable to the depth of the area in one case and strong motion artifacts in seven cases.

### 3.3. Inter-Reader Agreement

The inter-reader agreement for images obtained using DFI-EUS and for those obtained using CE-EUS is shown in Table 2. Both had kappa values that ranged from 0.6 to 0.8, thus demonstrating good inter-reader agreement.

The agreement rates among the three evaluators are shown in Table 3. The complete agreement rate for evaluations was 88% (53/60) with DFI-EUS and 83% (50/60) with CE-EUS. Incomplete agreement was shown in three positive cases and four negative cases of blood flow in DFI-EUS, while CE-EUS had six positive and four negative cases with incomplete agreement. No significant difference was observed in the complete agreement rates between DFI-EUS and CE-EUS (*p* = 0.60).

### 3.4. Diagnostic Accuracy of DFI-EUS

The diagnostic accuracy of DFI-EUS was evaluated based on 60 cases after excluding eight cases in which pancreatic blood flow was difficult to evaluate (Table 4). CE-EUS showed a contrast effect in the MNs of 24 cases (40%). Both CE-EUS and DFI-EUS identified blood flow signals in 20 cases (83%). Notably, none of the cases demonstrated positive DFI-EUS results in the absence of a contrast effect on CE-EUS. Based on the CE-EUS findings of the MNs, DFI-EUS demonstrated sensitivity of 83%, specificity of 100%, positive predictive value of 100%, negative predictive value of 90%, and accuracy of 93%. Among the 24 MNs with a contrast effect detected by CE-EUS, the abilities of CE-EUS and DFI-EUS to detect blood signals based on the MN size were compared (Table 5). Regarding MNs smaller than 5 mm, blood flow was detected in 7 cases of CE-EUS and 6 cases (86%) of DFI-EUS. For MNs measuring between 5 and 10 mm, blood flow was detected in 9 cases of CE-EUS, whereas it was detected in 6 cases (67%) of DFI-EUS. In contrast, for MNs larger than 10 mm, both modalities successfully detected blood flow in all 8 cases. A specific example of blood flow evaluation for MNs smaller than 10 mm is shown in Figure 4.

### 3.5. Long-Term Outcomes of Patients Diagnosed with Mucin Clots

Of the 36 cases diagnosed as mucus clots because of the absence of blood flow on DFI-EUS and CE-EUS, 33 were regularly followed at our institution. As of December 2024, the median follow-up period was 696 days (range: 168–1294 days), with 28 cases observed for over one year. No malignant findings were identified during follow-up.

### 3.6. DFI-EUS Evaluation of Surgically Resected Cases

Of the 68 cases, 17 with enhancing MNs larger than 5 mm on CE-EUS were classified as having high-risk stigmata. Among these, DFI-EUS identified linear or dotted signals within the MNs in 14 (82%) cases. Surgery was performed for 13 of these 17 cases (Figure 5). The final pathological diagnosis of the surgically resected cases included low-grade dysplasia for six cases and intraductal papillary mucinous carcinoma for seven cases. Among the seven intraductal papillary mucinous carcinoma cases, DFI-EUS detected dotted signals in two cases and linear signals in four cases; one case was difficult to evaluate. Similarly, among the six low-grade dysplasia cases, dotted signals were observed in two cases and linear signals were observed in three cases; one case was difficult to evaluate. No significant association between the pathological diagnosis and DFI signal findings was observed (*p* = 1.00).

## 4. Discussion

This study analyzed the ability of DFI-EUS to evaluate the blood flow of MNs in patients with IPMNs. Regarding the gold standard of blood flow, the number of cases with surgical specimens was limited, and the sensitivities of CE-CT and MRI for detecting MNs are insufficient. Furthermore, EUS-FNA has relatively low sensitivity for evaluating MNs [26]. Because of these limitations and the high accuracy for detecting MNs [19], CE-EUS was considered the most reliable method of assessing MNs. Based on the CE-EUS findings, the sensitivity, specificity, and accuracy of DFI-EUS for the detection of MN blood flow were 83%, 100%, and 93%, respectively.

DFI is a recently developed technique categorized as MVFI, which enables detailed evaluation of microvascular blood flow in ultrasonography [27]. Compared to conventional Doppler imaging, MVFI offers higher sensitivity, allowing the visualization of low-velocity blood flow and fine vascular structures [20]. This technology is particularly valuable for assessing tumor vascularity and detecting microvascular lesions. Numerous studies have reported the clinical utility of MVFI in ultrasonography [28,29,30,31,32,33,34,35,36,37,38,39,40,41]. The utility of MVFI has been reported for distinguishing between benign and malignant lesions in the liver, gallbladder, thyroid, parathyroid, kidney, breast, and lymph nodes, as well as for evaluating blood flow after aneurysm treatment and assessing the activity of inflammatory diseases. However, there have been no reports on the use of MVFI for the evaluation of pancreatic diseases in transabdominal ultrasonography. This is likely because visualizing the entire pancreas with transabdominal ultrasonography is challenging. The availability of DFI in EUS has been advancing the application of MVFI in areas accessible via EUS, such as the pancreas, gallbladder and gastrointestinal subepithelial lesions [22,23,24,42,43,44].

Yamashita et al. reported that DFI-EUS is more sensitive than eFLOW-EUS for identifying intratumoral vessels of gastrointestinal stromal tumors [22]. Regarding gallbladder lesions, the usefulness of DFI-EUS for differentiating gallbladder neoplasms from sludge has been reported [23]. Additionally, irregular vascular patterns have been associated with malignant gallbladder lesions. The sensitivity, specificity, and accuracy of DFI-EUS for detecting malignant gallbladder lesions are 89%, 100%, and 92%, respectively [23,42]. Yamashita et al. and Miwa et al. reported that DFI-EUS is also useful for the differential diagnosis of solid pancreatic lesions because it allows the visualization of precise vascular structures [43,44]. Miwa et al. reported that pancreatic cancer lesions are characterized by the presence of hypovascular vessels or dotted vessels around the tumor. They also demonstrated that the sensitivity and accuracy of DFI-EUS for detecting pancreatic cancer are 99% and 88%, respectively, which are higher than those of B-mode and eFLOW imaging. During the evaluation of MNs in IPMNs, DFI-EUS provides vessel detection superior to that of eFLOW-EUS [23], which is effective for differentiating MNs from mucus clots [24]. However, previous studies were limited because they included patients with other biliary and pancreatic diseases, resulting in a small number of IPMN cases. Therefore, this study evaluated DFI-EUS findings of MN blood flow based on CE-EUS findings using a larger number of cases.

Regarding contrast enhancement, international consensus guidelines have identified CE-CT and CE-MRI as the gold standards for diagnosing MNs in IPMNs. To objectively evaluate the blood flow of MNs, it is necessary to detect the presence of MN within pancreatic cysts or the main pancreatic duct using non-contrast imaging. However, CT and MRI often have insufficient spatial resolution for detecting MN due to the slice thickness, making blood flow evaluation difficult. In contrast, the guideline for IPMNs states that CE-EUS is effective in detecting MN in IPMN [5]. CE-EUS is considered superior to CE-CT for detecting MNs with contrast enhancement [17,19], and EUS is preferred by skilled endoscopists.

In this study, blood flow around the pancreas could not be evaluated on DFI-EUS in eight cases. The challenges in evaluation were due to deep lesions and significant motion artifacts. Compared to CE-EUS, DFI-EUS is influenced by the depth of lesion and motion artifacts. Theoretically, DFI depicts variations in the lesions; therefore, it strongly influences the depth of the lesion and ROI size. Lesions in deep areas and enlarged ROIs decrease the pulse repetition frequency and signal-to-noise ratio, which strongly influence the sensitivity of DFI-EUS. Therefore, in this study, the interpretation of images obtained using DFI-EUS was challenging when deep lesions and strong respiratory motions and vascular pulsations that caused motion artifacts were present.

There were four cases in which contrast enhancement was observed with CE-EUS; however, blood flow signals were not detected with DFI. Because of the small sample size, specific reasons for this technical failure to detect blood flow could not be identified. Regarding the ability to detect blood flow based on MN size, DFI-EUS successfully detected blood flow in all MNs larger than 10 mm. With DFI-EUS, some MNs smaller than 10 mm did not show blood flow. However, MNs smaller than 5 mm had a higher detection rate compared to those measuring 5–10 mm (86%; 6/7 cases vs. 67%; 6/9 cases). This suggests that in the four cases where blood flow was not detected with DFI-EUS, the depth of the lesions or motion artifacts, rather than MN size, may have affected detection. Among the skilled endoscopists with EUS experience, the kappa coefficients of the inter-reader agreement for DFI-EUS and CE-EUS were between 0.6 and 0.8, indicating good agreement. Furthermore, a high complete agreement rate of 88% was observed among the three evaluators. This high level of agreement was likely attributable to the clear evaluation criteria, which were limited by the presence or absence of blood flow. Interpretations of images obtained using CE-EUS were simple and focused on whether contrast agent perfusion was present in the MN. Similarly, interpretations of images obtained using DFI-EUS were straightforward and only required the identification of dotted or linear signals within the MN.

In surgically resected cases, an attempt was made to differentiate low-grade dysplasia from carcinoma using DFI-EUS findings. This approach was based on the assumption that dotted signals are more likely to reflect small and discrete blood vessels or areas with minimal vascularization, whereas linear signals are indicative of larger and more organized blood vessels. However, the appearance of these signals varies depending on the angle of the ultrasound probe, and linear signals sometimes appear dotted when observed from certain angles. Additionally, the small sample size posed a limitation to the analysis. Therefore, the findings of this study suggest that DFI-EUS is useful only for determining the presence or absence of blood flow in the MN. A unique advantage of DFI-EUS over CE-EUS is its ability to evaluate vascular structures, which has a possibility of differentiation between benign and malignant MNs based on vascular architecture in the future. Further evaluations with larger sample sizes are necessary to differentiate malignancy based on the vascular structure.

This study highlighted the significant advantages of DFI-EUS as a noninvasive and effective technique for evaluating MN blood flow. It also offers the advantage of enabling MN blood flow evaluation without contrast agents. However, its accuracy may be affected by factors such as lesion depth, respiratory movement, and arterial pulsation. In such cases, CE-EUS is useful as a complementary method to enhance the accuracy of MN blood flow evaluation. The sensitivity of DFI-EUS was high at 83%; however, among the 24 cases in which CE-EUS detected blood flow in MN, 4 cases were false negatives when assessed with DFI-EUS. Because DFI-EUS had a specificity of 100%, it would be preferable to first perform DFI-EUS and then confirm with CE-EUS in cases where the result is negative. This approach can reduce the need for contrast imaging, leading to shorter examination times and lower medical costs.

This study has several limitations. First, this was a single-center retrospective study with a small sample size and a small number of surgically resected cases. Second, this study focused solely on vascularity findings obtained through DFI-EUS. Consequently, laboratory data such as CA19-9 were not collected, and correlations between DFI-EUS findings and these laboratory parameters were not analyzed. Third, the image evaluation was based solely on still images that were retrospectively reviewed. The process of selecting these still images may have introduced selection bias and may not fully represent real-time image interpretation. Forth, because EUS image evaluations are typically performed using real-time videos, the reliance on static images might have resulted in assessment deviations compared to dynamic evaluations. Finally, during this study, CE-EUS was used as the gold standard to evaluate the capability of DFI-EUS to detect blood flow. Consequently, this study design did not allow for a statistical comparison of the diagnostic accuracy of these two modalities. To address these limitations, a prospective study with a larger number of pathologically confirmed cases and the inclusion of real-time video images is necessary to further clarify the diagnostic accuracy and potential clinical applications of DFI-EUS.

## 5. Conclusions

DFI-EUS can be used as an alternative to CE-EUS for IPMNs because of its simplicity and cost-effectiveness. DFI-EUS can detect blood flow in the MN and has the potential to become a useful tool for determining surgical indications for IPMNs.

## Figures and Tables

**Figure 1 diagnostics-15-00196-f001:**
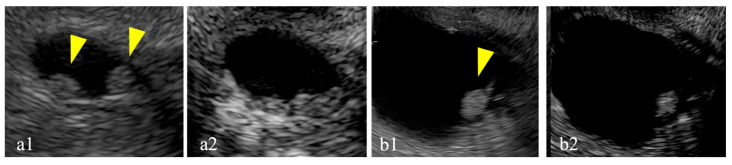
Mucus clots are difficult to differentiate from mural nodules. Case 1: a mural lesion in a cyst observed using conventional B-mode imaging (**a1**) exhibiting no enhancement (yellow arrow) (**a2**), thus leading to a diagnosis of a mucus clot. Case 2: a mural lesion in a cyst observed using conventional B-mode imaging (**b1**) exhibiting enhancement during imaging (yellow arrow) (**b2**), thus leading to a diagnosis of a mural nodule.

**Figure 2 diagnostics-15-00196-f002:**
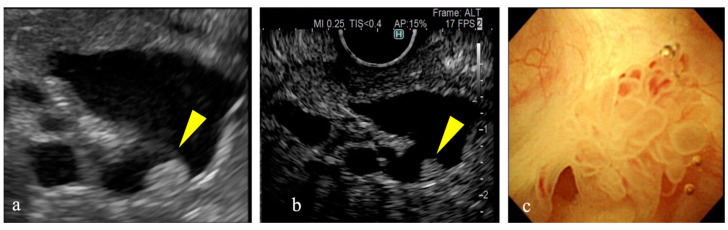
A mural nodule exhibiting enhancement. (**a**) Conventional B-mode imaging shows a hyperechoic mural nodule inside a cyst (yellow arrow). (**b**) A contrast-enhanced image shows homogeneous enhancement in the mural nodule (yellow arrow). (**c**) A papillary mural nodule is observed during peroral pancreatoscopy.

**Figure 3 diagnostics-15-00196-f003:**
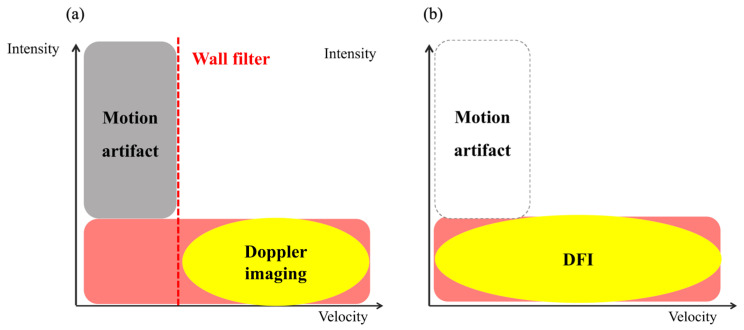
Schema of images obtained using Doppler imaging and detective flow imaging. (**a**) During Doppler imaging, low-velocity blood flow is not seen, and motion artifacts are observed near the wall filter. (**b**) Detective flow imaging uses an original algorithm to detect low-velocity vessels and reduces motion artifacts.

**Figure 4 diagnostics-15-00196-f004:**
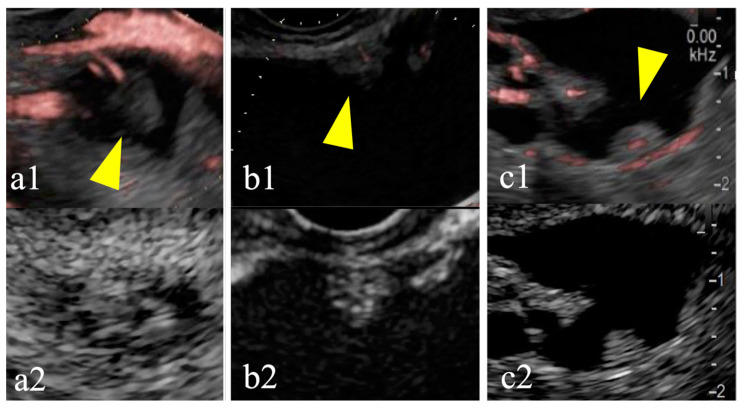
Detective flow imaging (DFI) and contrast-enhanced (CE) endoscopic ultrasonography (EUS) images of mural nodules smaller than 10 mm. Case 1 is observed using DFI-EUS (yellow arrow; **a1**) and CE-EUS (**a2**). Case 2 is observed using DFI-EUS (yellow arrow; **b1**) and CE-EUS (**b2**). Case 3 is observed using DFI-EUS (yellow arrow; **c1**) and CE-EUS (**c2**). DFI-EUS images of linear or dotted vessels inside the mural nodules. All CE-EUS images show homogeneous enhancement.

**Figure 5 diagnostics-15-00196-f005:**
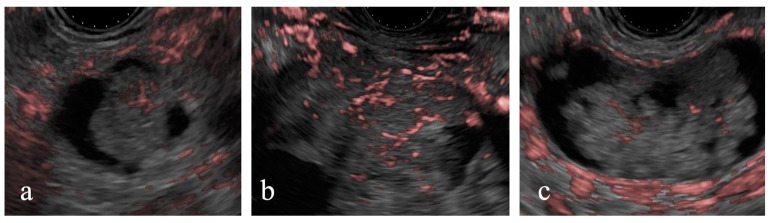
Detective flow imaging of mural nodules in surgically resected cases. Linear vessels are observed inside the mural nodule, intraductal papillary mucinous carcinoma (**a**,**b**), and low-grade dysplasia (**c**).

**Table 1 diagnostics-15-00196-t001:** Patient characteristics (*N* = 68).

Age, median (range), years	74 (50–87)
Male/female patients, *n* (%)	38/30 (56/44)
BD-IPMN/MD-IPMN/Mixed IPMN, *n* (%)	46/6/16 (68/9/23)
MPD diameter, median (range), mm	3.1 (1–23)
Maximum cyst diameter, median (range), mm	23 (6–76)
Location of MN, *n* (%)	Ph 40 (59)/Pb 16 (23)/Pt 12 (18)
Maximum MN diameter, median (range), mm	5.0 (1.7–30)

BD, branch duct; IPMN, intraductal papillary mucinous neoplasm; MD, main duct; MN, mural nodule; MPD, main pancreatic duct; Pb, pancreatic body; Ph, pancreatic head; Pt, pancreatic tail.

**Table 2 diagnostics-15-00196-t002:** Kappa coefficients for each reader combination.

Modality	Readers A and B	Readers B and C	Readers A and C
CE-EUS	0.75	0.79	0.79
DFI-EUS	0.75	0.65	0.73

CE-EUS, contrast-enhanced endoscopic ultrasonography; DFI-EUS, detective flow imaging endoscopic ultrasonography.

**Table 3 diagnostics-15-00196-t003:** MN blood flow assessment with DFI-EUS and CE-EUS.

		Blood Flow of the MN	
		Positive	Negative
DFI-EUS	Complete agreement	17	36
	Incomplete agreement	3	4
CE-EUS	Complete agreement	18	32
	Incomplete agreement	6	4

CE-EUS, contrast-enhanced endoscopic ultrasonography; DFI-EUS, detective flow imaging endoscopic ultrasonography; MN, mural nodule.

**Table 4 diagnostics-15-00196-t004:** Blood flow signal detection in MNs: DFI-EUS accuracy.

		CE-EUS	
		Positive	Negative
DFI-EUS	Positive	20	0
	Negative	4	36

CE-EUS, contrast-enhanced endoscopic ultrasonography; DFI-EUS, detective flow imaging endoscopic ultrasonography.

**Table 5 diagnostics-15-00196-t005:** MN size and blood signal detection.

*N* = 24	Positive Blood Signal	
	CE-EUS	DFI-EUS
MN < 5 mm	7	6
MN ≥ 5 but <10 mm	9	6
MN ≥ 10 mm	8	8

CE-EUS, contrast-enhanced endoscopic ultrasonography; DFI-EUS, detective flow imaging endoscopic ultrasonography; MN, mural nodule.

## Data Availability

The datasets generated during and/or analyzed during this study are available from the corresponding author upon reasonable request.

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
