# Peer review of "Diagnostic Accuracy of Detective Flow Imaging Endoscopic Ultrasonography for Evaluating Blood Flow Within Mural Nodules of Intraductal Papillary Mucinous Neoplasms"

_diagnostics, 2025, doi:10.3390/diagnostics15020196_

Round 1
Reviewer 1 Report
Comments and Suggestions for Authors
Dear All,
Thank you for allowing me to peer review this manuscript on the diagnostic accuracy of detective flow imaging (DFI) endoscopic ultrasound (EUS) in evaluating mural nodules within intraductal papillary mucinous neoplasms (IPMNs). In their well-designed single-center retrospective study, the authors demonstrated that EUS-DFI could represent a valid adjunctive tool in this context, with good specificity and positive predictive value. However, it does not exhibit sufficient sensitivity to replace contrast-enhanced EUS (CE-EUS), which remains the gold standard, particularly for smaller (<5 mm) mural nodules.
In my opinion, before this article can be considered for publication in Diagnostics, the following revisions are necessary:
- Page 1, Line 18: The abstract is too short. The authors should expand it by adding more detail to the materials and methods section and including more data in the results section.
- Page 3, Line 98: For the definition of mural nodules, do you also consider a dimensional range? Please clarify.
- Page 4, Line 137: First, add the initials of the experts who reviewed the images. Second, clarify the evaluation process: Did the same expert review the images of the same mural nodules, first evaluating CE-EUS images and then DFI-EUS images? If so, this could introduce bias, as the expert might be influenced by the CE-EUS image. Please address this potential bias.
- Page 5, Line 153: Add a clear definition of diagnostic accuracy.
- Page 5, Line 155: Did you also collect data on serum levels of Ca19.9? If so, did you observe any correlation between marker levels and the vascular pattern of the nodules observed with DFI-EUS?
- Page 6, Line 203: Clarify whether there was a significant statistical difference in technical success between DFI-EUS and CE-EUS.
- Page 8, Line 281: When referring to “adenoma,” do you mean low-grade or high-grade dysplasia? Please specify.
Reviewer 2 Report
Comments and Suggestions for Authors
Thank you for allowing me to review your manuscript. Please find my comments below.
Summary
This is a retrospective study which evaluated the diagnostic accuracy of the Defective Flow imaging EUS imaging technology to accurately identify mural nodules in patients with presumed IPMN, pancreatic cystic lesions. All the patients with IPMNs who underwent contrast enhanced EUS and DFI EUS were included. The images obtained during the procedure were retrospectively reviewed to identify vascular activity in the confirm mural nodule from mucus clot. The study found that compared to CE-EUS DFI-EUS demonstrated a sensitivity, specificity , and accuracy of 83%, 100% and 93%, respectively, for detecting the mural nodule blood flow. They concluded that the DFI-EUS imaging was a simpler, cost-effective and less time-consuming alternative to CE- EUS to identify mural nodules in patients with PMN.
Originality
This is an original study without any previous studies discussing this topic.
Weaknesses and comments are as below.
General comments
1. Please clarify how all the lesions were identified to be IPMNs? If they were presumed, please mention that in the abstract and the methods.
2. Please add a definition of mural nodule with reference.
3. Please mention other ways to distinguish mural nodules from a mucus clot on EUS.
4. Please briefly mention DFI technology.
5. Please clarify what “rating” system was used by the three readers?
6. Please mention the significance of the dotted and linear signals seen on EUS.
7. CE-EUS was used as the gold standard?
Round 2
Reviewer 1 Report
Comments and Suggestions for Authors
Dear All,
The authors have made all requested corrections and the paper can now be considered for publication.